# Using near-surface atmospheric measurements as a proxy for quantifying field-scale soil gas flux

Andrew Barkwith[1], Stan E. Beaubien[2], Thomas Barlow[1], Karen Kirk[1], Thomas R. Lister[1], Maria C. Tartarello[2] and Helen Taylor-Curran[1]

[1]British Geological Survey, Environmental Science Centre, Nottingham, NG12 5GG, UK.

[2]Dipartimento di Scienze della Terra, Università di Roma "La Sapienza", Rome, 00185, ITALY.

*Correspondence to*: Andrew Barkwith (andr3@bgs.ac.uk)

**Abstract.** We present a new method for deriving surface soil gas flux at the field-scale, which is less fieldwork intensive than traditional chamber techniques and less expensive than those derived from airborne or space surveys. The 'open-field' technique uses aspects of chamber and micrometeorological methods combined with a mobile platform and GPS to rapidly derive soil gas fluxes at the field-scale. There are several assumptions in using this method, which will be most accurate under stable atmospheric conditions with little horizontal wind flow. Results show that soil gas fluxes, when averaged across a field site, are highly comparable between the open-field method and traditional chamber acquisition techniques. Atmospheric dilution is found to reduce the range of flux values under the open-field method, when compared to chamber derived results at the field-scale. Under ideal atmospheric conditions it may be possible to use the open-field method to derive soil gas flux at an individual point, however this requires further investigation. The open-field method for deriving soil-atmosphere gas exchange at the field-scale could be useful for a number of applications including quantification of leakage from $CO_2$ geological storage sites, diffuse degassing in volcanic and geothermal areas and greenhouse-gas emissions, particularly when combined with traditional techniques.

## 1 Introduction

The study of soil-atmosphere gas exchange has become more prominent over the past couple of decades. Objectives for these studies are wide-ranging, for example: the study of volcanic degassing (Carn et al., 2016; Cardellini et al., 2017); quantification of carbon budgets (Houghton and Nassikas, 2017; Le Quéré et al., 2017); greenhouse gas (GHG) emission studies (Oertel et al., 2016); and identifying potential leakage from Enhanced Oil Recovery (EOR) and Carbon Capture and Storage (CCS) sites (Korre et al., 2011; Beaubien et al., 2013; Jones et al., 2014). Soil gas emissions are directly measured at points or spots using chamber techniques (Pumpanen et al., 2004) or over restricted areas through micrometeorological methods (Dugas, 1993). At regional and national scales, airborne and space measurements are used to derive soil gas emissions using empirical and process-oriented models for post-processing. These regional scale methods lack detail required for field-scale studies ($10^1$ to

$10^3$ m$^2$) and may be prohibitively expensive (Oertel et al., 2016). Feitz et al. (2018) provide a comparison of many of these techniques under a controlled gas release.

Closed loop flux chamber-based analyses utilize an open-bottomed chamber with a known footprint and volume placed on the soil surface, allowing gases emitted by the soil to accumulate within the chamber headspace (Rolston, 1986). From analysis

of the gas mixing ratios within the chamber over time, the flux of gas from the soil can be derived for that small spot of the land surface. In contrast, an open loop technique passes air through the sample chamber at a known flow rate, until a steady-state concentration is observed, from which a flux rate is derived (Denmead, 2008). Both techniques require measurements at a large number of points to estimate field-scale fluxes via interpolation, with the caveats that sample density is sufficient to represent site spatial variability and that flux is static with respect to time during the measurement period (Gao et al., 1998).

The use of chambers to represent soil gas fluxes at field-scale is often highly time-intensive and prone to interpolation-related uncertainty (Elío et al., 2016), particularly for sites with heterogeneous soils or geology.

Micrometeorological methods use eddy covariance (EC) techniques to derive soil gas flux. A three-dimensional sonic anemometer is coupled to a gas analyser attached to a tower or mast, allowing measurements that incorporate areas up to

several square kilometres under the right atmospheric and terrain conditions (Myklebust et al., 2008). Continuous EC measurements over a period of time (usually days to weeks) allow soil gas fluxes for a particular parcel of land (the EC footprint) to be derived from absolute gas concentrations, temperature, and vertical and horizontal wind flows (see for example, Eugster and Merbold, 2015). The EC footprint location and size are calculated through post-processing of the high-resolution data, averaged over longer time intervals (Aubinet et al., 2012). EC methods require a fully turbulent flux, where the majority

of vertical movement is driven by eddies, and uniform, homogeneous terrain, where air density fluctuations and convergence/divergence are negligible (Lee et al, 2004). Soil gas flux from EC is derived by integrating the net fluxes upwind from the measurement point (Eugster and Merbold, 2015). A key component of the EC method is calculating (a posteriori) the pathway from the instrument sensors to the soil surface under turbulent conditions, which leads to multiple assumptions (see Baldocchi, and Meyers, 1998). Tower or tripod based EC methods are difficult to utilise for consistent identification of soil

gas flux at any particular location, as they are reliant on wind direction, surface roughness and atmospheric conditions to determine the location of their footprint. Roving or mobile EC towers can effectively enlarge the EC footprint to cover any particular location, however these techniques take days to weeks of measurement to provide sufficient coverage at the field-scale (Eugster et al., 1997; Billesbach et al., 2004).

Unmanned Aerial Vehicles (UAVs) could potentially be used to capture field-scale flux in the future, but currently have limited flight times (<30 minutes) and weight restrictions that limit sensor options to small, low-power devices (see for example, Danilov et al., 2015; Hass et al., 2014). These low-power sensors do not currently have the sensitivity to observe small flux

anomalies at flight height and speeds of fixed-wing UAVs, and slower copter-style UAVs generate too much downdraft for an accurate measurement (Li et al., 2020).


There is currently a lack of practical, fast, inexpensive methods for quantifying soil gas flux at field-scales, which are highly relevant to leakage and degassing studies. The objective of this paper is present a new 'open-field' method that uses aspects of chamber and micrometeorological methods combined with a mobile platform and GPS to rapidly derive soil gas fluxes at the field-scale. We assess this method against traditional chamber techniques for field locations within the UK and Italy, and

discuss the explicit and implicit assumptions inherent in the presented technique.

## 2 Materials and methods

Development of a new field-scale soil $CO_2$ flux quantification method was focused on creating a mobile tool that could easily and quickly make measurements around a field site without the need for stopping at individual locations, and that was valid on sloping or heterogeneous terrain. Here we describe the theoretical aspects, assumptions made, components used to

undertake the measurements and post-processing requirements.

### 2.1 Experimental theory

As we approach the ground surface, frictional drag reduces horizontal wind speed to near-zero. The depth of this frictional influence depends on the roughness of the ground (Oke, 1987). By assuming that there is no horizontal wind flow close to the surface, we can discretize the near-surface atmosphere into non-interacting boxes of air, each with a base fixed on the ground

surface, and treat each of these as a type of 'open' dynamic flux chamber. Open chambers use two openings, an inlet that draws ambient air, and an outlet to generate a continuous gas flow. The gas flux is calculated by the concentration difference between these two ends under a known flow rate through the system (Kutsch et al., 2009). As such, if we know the concentration of a particular gas within our air boxes, the atmospheric background concentration and the vertical flow rate of air up through the box, we can calculate soil gas flux in a similar way. In other words, we calculate the amount of extra gas required to maintain

a particular stable concentration near the surface. This may be derived from the Ideal Gas Law:

$$F = M_m \left( \frac{PVw}{RT} \right) \tag{1}$$

$$V = (c_O - c_B).10^{-6} \tag{2}$$


Where flux, $F$ (grams per square metre per second), is calculated using: the atmospheric pressure, $P$ (Pascal); the ideal gas constant, $R$ (8.31446 cubic metres per Pascal per Kelvin per mole); temperature, $T$ (Kelvin); the molar mass of the gas being sampled, $Mm$, (grams per mole); the vertical wind speed, $w$ (metres per second); and $V$ (cubic metres per cubic metre), the

volume of gas (cubic metres) occupied by the difference between observed ($c_O$) and background ($c_B$) gas concentration (parts per million volume), per cubic metre of air. From the perspective of EC theory, this is similar to moving sensors from >2 m (standard for EC flux measurements) to ground level and reducing the footprint area to zero. Under this setup, the atmospheric effects on the pathway between source and EC sensors become negligible and we can dismiss the assumptions associated with turbulence and field properties.

The physical basis for this calculation can be described using a thought experiment. Suppose we have a box of air at the ground surface with a known, uniform gas concentration. As we know the volume of the box and the gas concentration, we know the weight of that gas within the box from the ideal gas law. Assuming the box is fully mixed, if we remove a known volume of gas from the top of the box, we can calculate the weight of gas removed for a given area of land per time-step. If we replace the displaced volume with background (external) air, there will be a change in the gas weight in the box (unless background and observed concentrations are equal) that equates to the weight of gas either added or removed at the soil surface for a given area of land per time-step, i.e., a soil gas flux.

## 2.2 Assumptions

There are several implicit and explicit assumptions in the experimental theory presented for deriving open-field soil gas fluxes. No horizontal wind flow at the measurement height is a major assumption and in practice does not hold true under certain conditions, particularly under high winds or on very smooth (aerodynamically) land surfaces. This can be tested in the field by measuring horizontal wind flow at or near to measurement height, which itself is related to the roughness length of the surface and meteorological conditions. Where horizontal wind flow is non-zero, measurements above a specific location are no longer spatially coherent with the ground directly below. Under these conditions, only an average open-field soil gas flux covering a particular area can be derived. The relationship between wind speed, aerodynamic roughness and the area of land required to gain a representative averaged soil gas flux is unknown, but it is likely to be similar to the derivation of flux footprint from the Eddy Covariance method (see for example, Horst (1999)). Without horizontal wind flow, there are no turbulent conditions to create the vertical wind components, however, even under moderately convective conditions, the vertical wind field is directly coupled to the temperature field through buoyant forces (Nilsson et al., 2012).

It assumed that a measurement point represents the entire box of air and that the air is fully mixed. As the experimental theory is scalable, the size of the box can be reduced to near-zero and, therefore, the assumption holds true. How well that measurement represents surrounding areas when interpolation is applied in post-processing is unknown. The same issue is faced by traditional chamber methods, however the open-field method results in a much higher density of measurements and thus a comparatively reduced uncertainty.

Air is assumed to be non-compressible and at a uniform temperature and density. The former is a standard assumption in atmospheric sciences and would require complex adjustments to calculate, however, given the scalability, the impact of compressibility differences would be minimal at a near-zero box volume. Temperature and pressure differences are accounted for in the calculation of gas weights using the Ideal Gas law.


Finally, we assume that replacement air comes from either background atmospheric or the soil surface. In reality, there will be some replacement from the surrounding air, which isn't necessarily at background. The greatest impact of this assumption occurs under atmospheric conditions that create high vertical wind speeds and, therefore, are likely to 'draw' air from the surrounding area, such as when the land surface is much warmer than the surrounding atmosphere. The impact of air-box

interaction, in comparison to chamber methods, should result in a smoother, less peaky dataset than that derived from chamber methods.

### 2.3 Field measurements and post processing

To gather the field-scale flux data, several instruments were mounted on a light-weight metal handcart which was pulled around the various field sites. To measure gas concentrations at the required short time intervals, we used either open path

lasers (Boreal Gasfinder3) or a gas analyser (Los Gatos Research Greenhouse Gas Analyser) to measure $CO_2$ at 1 Hz. These were mounted on the handcart and were sampling at a height of 10 cm from the ground surface. Vertical wind flow was measured at 10 Hz using a tri-axis sonic anemometer (Gill Windmaster) mounted at a fixed point in the field. Finally, a global positioning system (GPS) receiver was added to the cart to provide positional data for the gas and flow measurements. Figure 1 shows a prototype setup, with a sonic anemometer, open path laser systems and GPS fitted to a custom-built handcart. For

each of the field sites, the cart was pushed at a slow walking pace (~2 km h$^{-1}$) in a grid pattern. The timestamps for all instruments were synchronised at the start of the day and checked periodically for discrepancies. As the field data was collected for multiple research purposes (for example, leakage detection), the cart was sometimes returned to points with high gas concentrations to map specific areas in greater detail.

For comparison, traditional closed loop chamber methods were used to measure soil $CO_2$ flux on a regular grid, where possible, for each of the field sites. For practical purposes, grid spacing for the chamber measurements was determined by the size of the field and the time available to take samples; a total of 80 and 32 points were measured at the Italian and UK sites, respectively. An overview describing aspects of traditional chamber, EC and the open-field methods is given in Table 1.

Following collection of field data, time-series of observational datasets (GPS, meteorological and gas concentration) were used (by an algorithm written in C++ and using Eqn. 1 and 2) to derive the open-field soil gas flux at 1 Hz. Data from the sonic anemometer was averaged from 10Hz to the mid-point of each second. The GPS data allows the location of each data-point to

be logged and the derived soil gas flux was spatially interpolated between points using a standard kriging method. This interpolated dataset is used for the comparison of traditional chamber and open-field methods in Section 3.

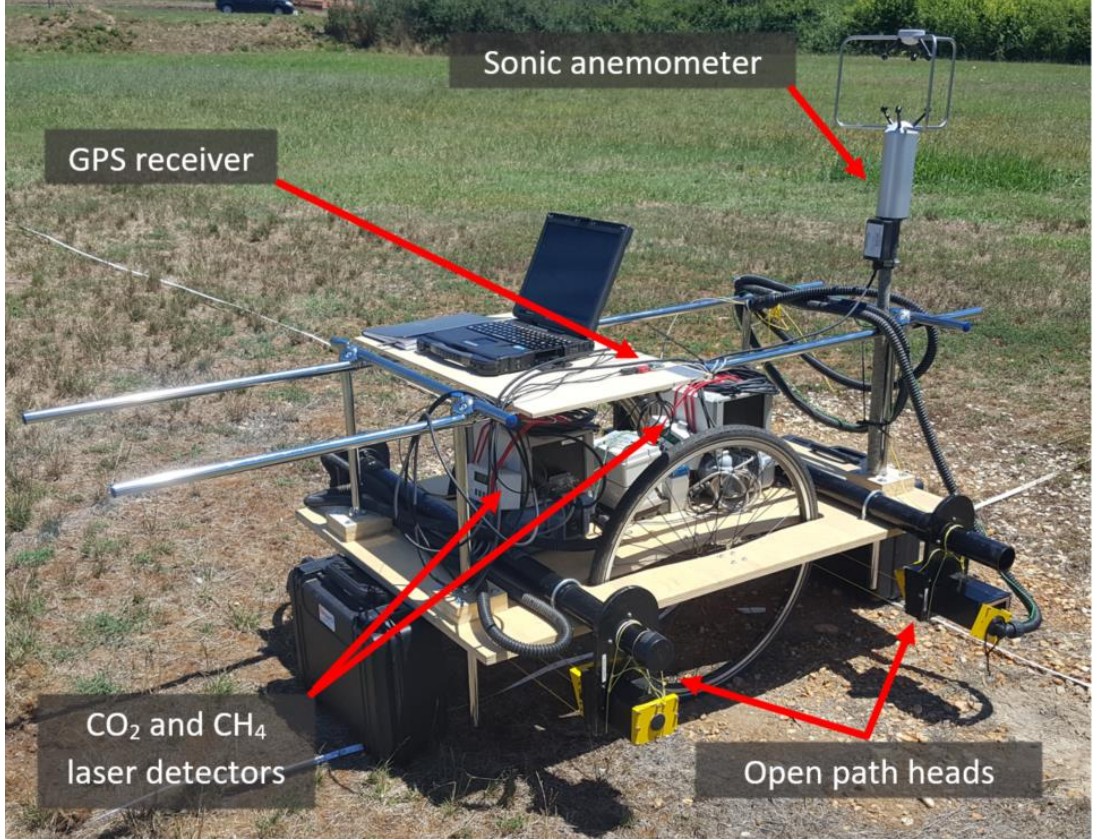

**Figure 1: Prototype of system used for this study. The system used to collect data for this research had the sonic anemometer located in a fixed position in the field (instead of cart mounted) and the CH4 open path laser replaced with a Los Gatos Research Greenhouse Gas Analyser. The cart was also replace with a lightweight metal cart that was easier to transport.**

**Table 1: Overview of the major characteristics of traditional chamber systems, the eddy covariance method and the open-field method to measure soil gas fluxes. Adapted from Eugster and Merbold (2015).**

| Aspect | Traditional Chambers | Eddy Covariance | Open-field |
|---|---|---|---|
| Spatial coverage | Small: few cm2 per chamber; Moderate: can interpolate between multiple measurements | Large: few m2 (bare soil) to several ha (tall forest), dependent on surface roughness and atmospheric conditions | Large: few m2 to several ha; limited by the speed at which the cart is pushed |
| Measurement time at field-scale | Moderate: hours to days depending on measurement spacing. | High: days to weeks depending on atmospheric conditions | Low: minutes to hours |
| Measurement type | Indirect: flux is calculated via the concentration increase over time during chamber closure | Direct: flux is measured as the covariance of changes in turbulence and gas concentration | Indirect: flux is calculated via the concentration difference to background and vertical components of wind |
| Instrument costs | Moderate: for manual chambers and analysis of the gas sample via gas chromatography; Moderate/high: for automatic chambers which are either connected to a gas chromatograph or a gas analyzer (e.g., infrared gas analyzer or laser absorption spectrometer) | Moderate: for the scaffolding or a tripod; High: for instruments capable of measuring turbulence (sonic anemometers) and gas concentrations (infrared gas analyzers, laser absorption spectrometers) at high temporal resolution (typically 20 Hz) | Low: for cart. High: for instruments capable of measuring turbulence (sonic anemometers) and gas concentrations (infrared gas analyzers, laser absorption spectrometers) at moderate temporal resolution (typically 1 Hz) |
| Maintenance costs (technical) | Low: for manual chambers; moderate: for automatic chambers as well as for carrier gases, for example, within a gas chromatography setup | Moderate: for replacing small technical devices and calibration gases; high: in the case of sensor replacement | Moderate: for replacing small technical devices and calibration gases; high: in the case of sensor replacement |
| Maintenance costs (labour) | High: due to length of time required for sample collection | Moderate: due to remote maintenance and less field activities | Low: due to length of time required for sample collection |
| Computing requirements | Low: flux calculation is based on few data points and can be script based | High: due to high-frequency data (> 10 Hz) and often data covering > 1 year | Moderate: high-frequency data over a short period and is script-based |


## 2.4 Study sites

To test and develop the new open-field technique two sites were chosen which have markedly different characteristics in terms of $CO_2$ flux origins and rates. The first is located in a mountainous valley near the small town of Ailano, Italy, situated about 150 km SE of Rome. This site consists of numerous flat agricultural fields where deep-origin, geologically produced $CO_2$ is

migrating towards the surface and leaking to the atmosphere from a large number of variably sized "gas vents" (Ascione et al., 2018). These gas vents, some of which are isolated while others overlap and merge, range in $CO_2$ flux rates that are slightly above the normal biological value of around 20 g $m^{-2}$ $d^{-1}$ to over 5,000 g $m^{-2}$ $d^{-1}$, with average values typically less than 300 g $m^{-2}$ $d^{-1}$. The second site, Sutton Bonington, UK, is home to the GeoEnergy Test Bed (GTB), a research facility that enables development and testing of innovative monitoring technologies to improve our understanding of impacts and processes in the

shallow subsurface. This site consists of relatively flat agricultural fields overlaying river terrace deposits and sandstone and mudstone formations. The data for this study was collected prior to any experimentation at the GTB as part of a baseline survey. As data from the Ailano and Sutton Bonington sites may be sensitive, exact locations are not given.

## 3 Results and discussion

Figure 2 shows $CO_2$ flux data (g m$^{-2}$ d$^{-1}$, note the difference in temporal units compared to the flux equation presented in the
methods section) from a single field at the Ailano site, with fluxes from the chamber method plotted on top of the interpolated
field-scale flux distribution. A visual comparison shows that the open field-scale method produces values that are of a similar
order of magnitude to those obtained by chamber methods, although there are clear differences at individual observation
locations. $CO_2$ flux data for each individual survey point at both field locations are plotted in Fig. 3.

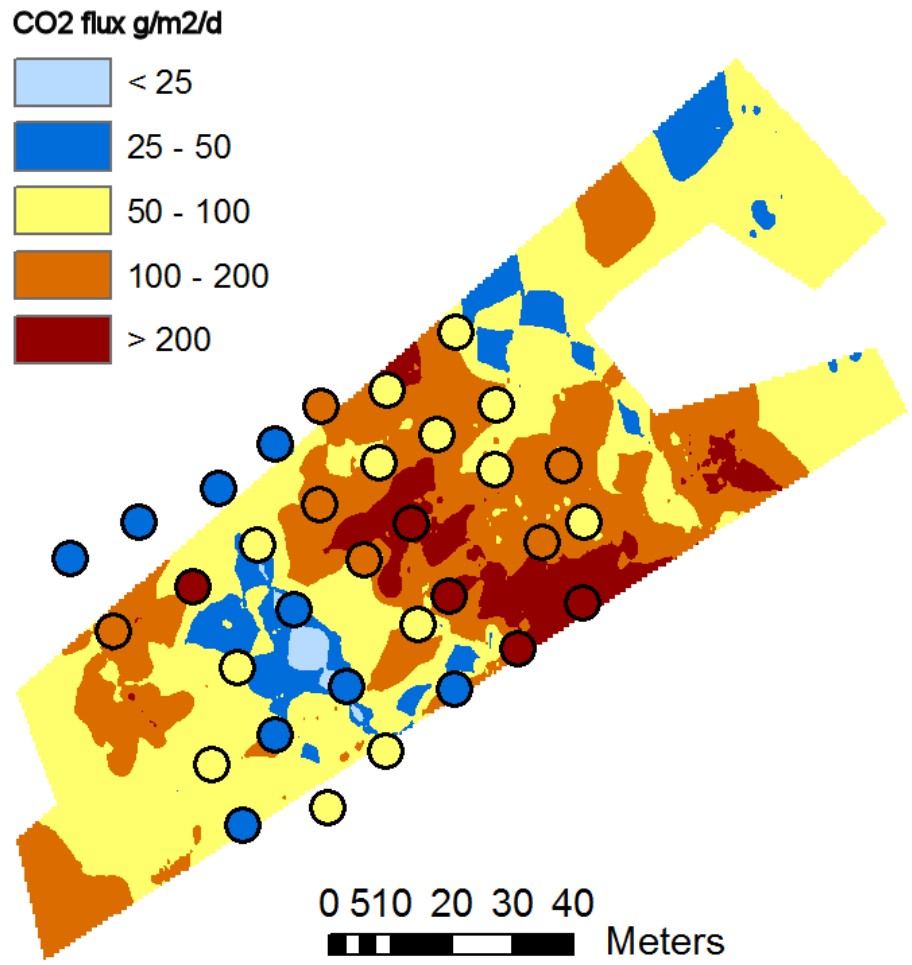


**Figure 2: $CO_2$ flux (g m$^{-2}$ d$^{-1}$) measured using a closed loop chamber technique (circles) and that derived using the open field-scale method (interpolated underlying plot), for one of the field sites in Ailano, Italy.**

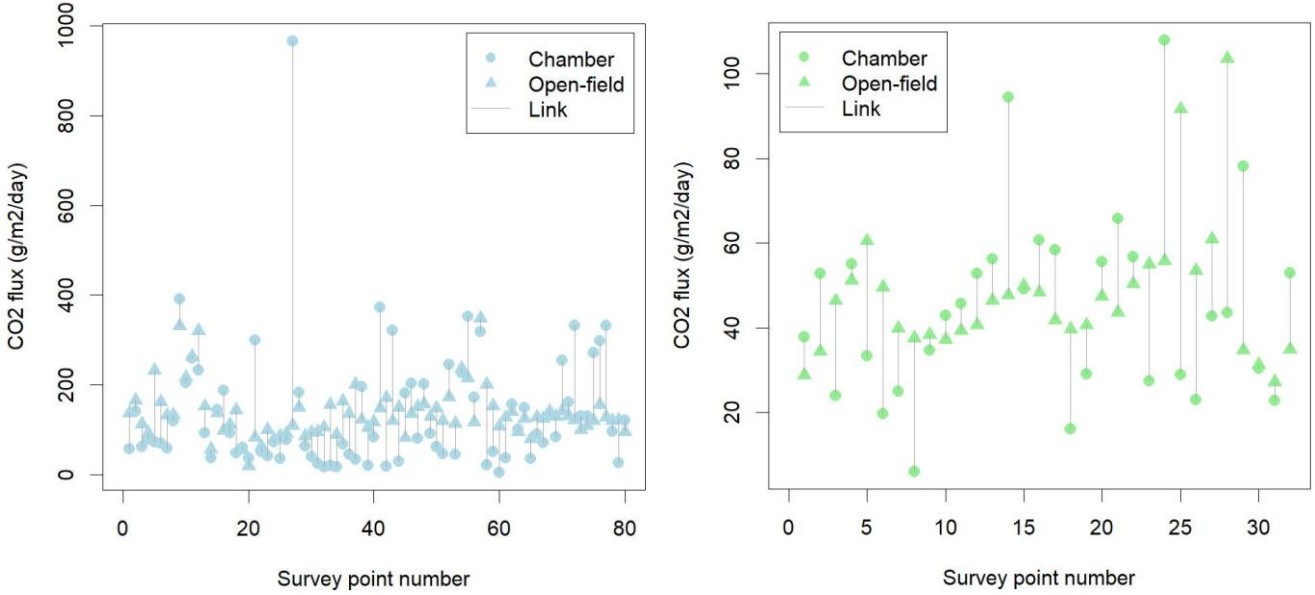

**Figure 3: Plots comparing CO₂ flux measured using the traditional chamber technique (circles) to those derived from the open-field method (triangles) for Ailano, Italy (blue) and Sutton Bonington, UK (green); note the much lower values at the latter. Link lines (grey) have been added for each survey point to aid in visual assessment.**

To quantitatively compare open field-scale and closed loop chamber methods at the larger scale we have de-localised the datasets and derived a set of summary statistics. The difference in mean $CO_2$ flux values between the techniques is 0.5 g m$^{-2}$ d$^{-1}$ for Ailano and 2.4 g m$^{-2}$ d$^{-1}$ for Sutton Bonington. Figure 4 shows box and whisker plots representing the rest of the summary statistics for both sites. The median for both techniques (50% percentile) is highly comparable for the two sites, while the open field-scale method exhibits less range between the 25% and 75% percentiles. Regression analysis (Fig. 5) shows the deviation from a 1:1 relationship between the point and mobile flux techniques. The coefficient of determination ($r^2$) between the techniques is 0.29 for Ailano and 0.08 for Sutton Bonington.

The results show that the average (mean and median) $CO_2$ flux obtained using the chamber technique and those derived from the open field-scale method presented in this study are highly comparable at field-study scales. The range (absolute and between quartiles) of $CO_2$ flux is smaller in the latter dataset, which, as discussed in the assumptions section, is likely due to atmospheric dilution.

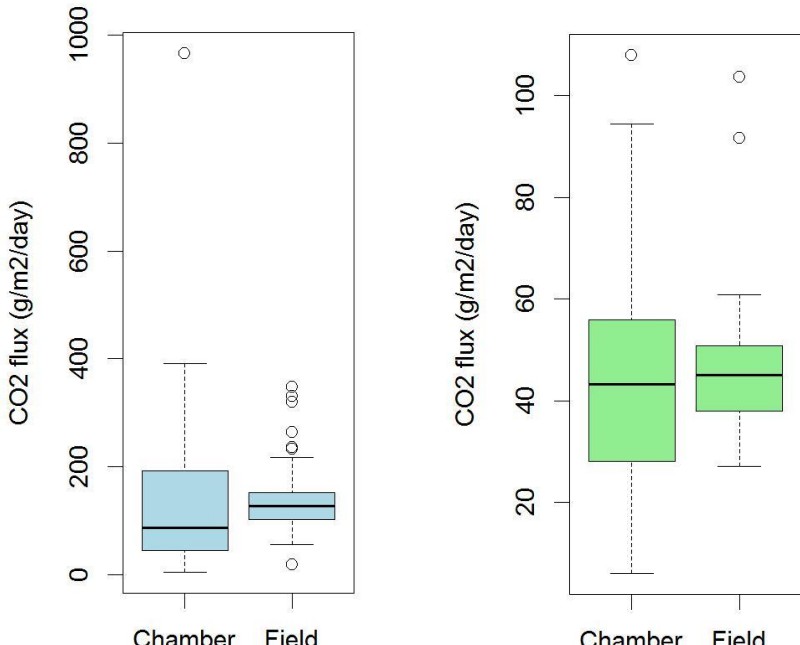

**Figure 4: Box and whisker plots comparing CO₂ flux measured using the traditional chamber technique to those derived from the field method for Ailano, Italy (blue) and Sutton Bonington, UK (green); note the much lower flux values at the latter. The boxes represent the 25% (bottom) and 75% (top) quartiles, and the central line the median. The whiskers extend to 1.5 times the inter-quartile range and outliers are given as individual points.**

215

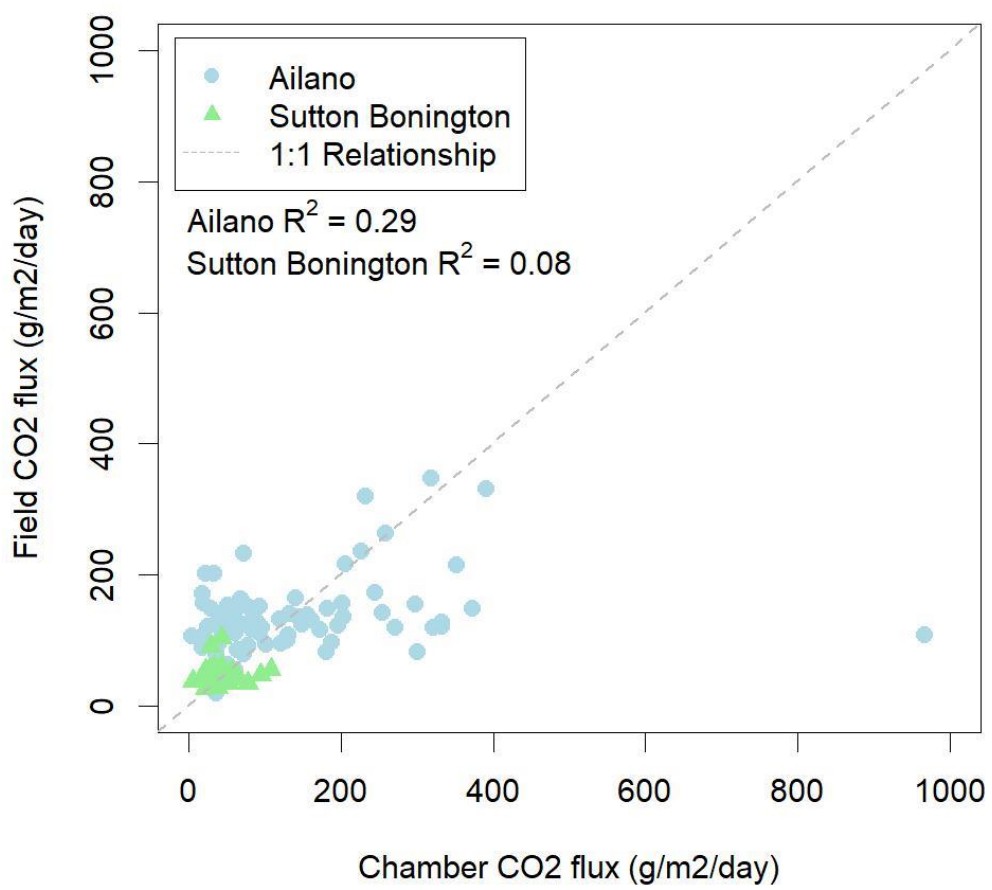

**Figure 5: CO₂ flux comparison of chamber vs open field-scale technique for the measurement sites in Ailano, Italy (blue) and Sutton Bonington, UK (green).**

220

At individual measurement locations there is the potential for correlation between the chamber and open-field derived techniques, as shown at the Ailano site. However, this was not apparent in the data collected for Sutton Bonington. This could be due to atmospheric conditions or that the observed flux values were much smaller at the UK site. We assessed wind speed and direction against the difference in flux between the two techniques, with low regression values indicating no relationships

225   between the wind variables and differences in flux (see supplementary materials).

## 4 Conclusions

The present work describes the theoretical basis and preliminary test results of a new method for rapidly estimating $CO_2$ flux from the ground surface over large areas, opening the door for future development and improvement of this hybrid approach. The developed "open-field" method, which combines aspects of open chamber and micrometeorological methods on a mobile platform, is less fieldwork intensive than traditional chamber techniques and cheaper than those derived from airborne or space surveys. Due to several assumptions, the most accurate results are expected under stable atmospheric conditions, with little horizontal wind flow. When derived soil gas fluxes are averaged at the field-scale, they are highly comparable to results obtained using traditional chamber techniques. As expected, atmospheric dilution leads to a reduced range of flux values under the open field-scale method. Under ideal atmospheric conditions it may be possible to use the new method to derive soil gas flux at an individual point, however this requires further investigation. The presented method of deriving soil-atmosphere gas exchange at the field-scale could be useful for a number of applications including leakage, degassing and greenhouse-gas emission studies. The results presented for the open-field flux method are limited in scope and it is recognised that further research is required to assess robustness under different environmental and meteorological conditions.

**Data availability**

The data that support the findings will be available from the National Geoscience Data Centre (https://www.bgs.ac.uk/geological-data/national-geoscience-data-centre/) following a 12 month embargo from the date of publication. An embargo is in place as the data are still being actively processed and interpreted within ongoing research projects.

**Acknowledgement**

The authors would also like to thank Doug Smith (Temporary Researcher, BGS) for useful discussions around deriving the flux equation. This work is published with permission of the Executive Directors of BGS. This publication received funding from the European Union's Horizon 2020 research and innovation programme under grant agreement No 653718 (ENOS project) and from the Natural Environmental Research Council under UKRI.

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
