# Peer review of "Using near-surface atmospheric measurements as a proxy for quantifying field-scale soil gas flux"

_Geoscientific Instrumentation, Methods and Data Systems, 2020_

## Referee Comment (RC1) · Anonymous Referee #1 · 5 May 2020

General comments. This is a very short (too short?) but interesting paper in the scope of GI. It presents a method based on the theoretical principles of flux measurements at the soil/atmosphere interface coupled with data acquisitions based on the principles of an Eddy Covariance system. In general, micrometeorological systems are operated at a fixed location, at some meters above the ground, to get information on gas emissions over a long period. Information is thus mostly dependent from wind direction among other variables. Here, the authors use the basics of EC but operated from a mobile platform on which gas sensors are mounted to allow gas monitoring at 10 cm above the ground. The flux is derived from these gas concentrations measurements by using the Ideal Gas Law and by discretizing the air layer into virtual boxes opened at top and

bottom, the vertical flow rate inside the box being determined using the anemometer from the EC system. This is an interesting approach, which is believed to benefit from a quite high rate of geographical coverage and from good location and description of gas emissions. Nevertheless, some important information is lacking in the manuscript; some comments are given below.

Specific comments. Introduction section: the authors compare the labor intensive ground-based method (flux measurements), the EC approach and airborne/space measurements (costly). Nowadays, there are several research institutes/companies worldwide developing drone-based solutions, using commercial or homemade sensor technologies, including open-path lasers, directed lasers, IR sensors... Some developments about this approach are welcome in the introduction, because, depending on the topography/vegetation, drone flights can also be operated at short distances from the ground, offering a geographical coverage rate far higher than the one reached by systems operated at walking speed. The reference Feitz et al. (2018) is mentioned but not as a basis of comparison e.g. for evaluating positive aspects and drawbacks. As noticed by the authors, there are some assumptions in the proposed approach. One of these assumptions is the absence of horizontal wind at land surface or at list little horizontal wind flow. More developments on this point are required to assess the potential use of the proposed method, and also on the related parameters such as soil roughness (e.g. Giannico et al., 2018. Contributions of landscape heterogeneity within the footprint of eddy-covariance towers to flux measurements. https://doi.org/10.1016/j.agrformet.2018.06.004). First, literature data are welcomed to evaluate the soundness of this major hypothesis. Second, data related to the acquisitions performed in Italy and UK have to be presented: there is no Figure showing the wind conditions during acquisitions and this information is lacking. This may partly explain some of the differences reported in Figure 1 (Italian site) and probably some of the differences observed at the UK site (data not shown but this can be deduced from Figure 3). On the contrary, are there some information on the diurnal variability of $CO_2$ emissions at the UK site that can explain, at least partly, the poor agreement between

chamber measurements and measurements using the authors' approach (chamber data were probably acquired over a longer period than the other data)? Do the authors think the "open-field scale" approach can be used with a sufficient degree of confidence at the UK site or would they prefer using the chamber measurements? Back on the "absence of horizontal wind" assumption: the authors mention potential applications of their method, including leakage detection. What about leakage detection when there is little to no vegetation on the ground (the CO2FieldLab experiment is mentioned in the references)? Does the assumption seem realistic in that case? What about using such a method in desert environments? The authors also mention the Weyburn case: what could be the influence of frozen conditions on wind conditions close to ground surface? The acquisitions were performed with a sensor mounted at 10 cm from ground surface: if the vegetation is higher than 10 cm and not grazed or mown (spring/summer conditions), how can be the method adapted? A picture showing/describing the "open-field scale" system in field-use conditions is lacking. It is always informative to have such Figure. Technical approach: it would have been interesting to compare with a third approach, intrinsically related to the "open-field scale" approach: the "traditional" EC monitoring. Why has this not been performed? It would have given interesting information on the benefits of the "open-field scale" approach, e.g.: is the "open-field scale" approach offering the same smoothing of anomalies than the EC approach, or does it give a better rendering? On the discrepancies between chamber data and "open-field scale" approach (especially for the UK site): a Figure comparing the results of the two methods is missing and this is needed; Figure 3 is not sufficient because we only see a point cloud. There is nearly no discussion on the differences between the two approaches for the UK site (only line 159). There is a lack of comparison with literature data, even if the "open-field scale" approach is not described in there because of its novelty. For example, what can be the differences with the use of vehicle-mounted (or walking use) of open-path lasers without quantifying the vertical wind flows? Because the "open-field scale" approach, like the EC approach, is supposed to give smoothed information on extreme values, what is the monitoring approach suggested by the authors in case there is a need to quantify these very high values. Use of "open-field scale" approach first and then perform chamber measurements, e.g. as suggested by Eugster and Merbold, 2015 (Eddy covariance for quantifying trace gas fluxes from soils. SOIL 1, 187–205. https://doi.org/10.5194/soil-1-187-2015)?

Technical corrections: none (well written)

---

## Referee Comment (RC2) · Anonymous Referee #2 · 19 Jun 2020

General Comments The authors describe a new approach to measure soil-atmosphere fluxes on the field scale which might overcome the limitations of traditional chamber techniques (precise, but only laborious repeated measurements allow to get map an area) and traditional micrometeorological measurements (fixed location, no spatial resolution of the footprint). The approach is interesting, but it lacks many details that would be necessary to get a better idea if this approach really holds the potential to fill the gap of field scale measurements. The idea is intriguing to put a kind of Eddy -Covariance system on a small cart in order to get more spatial resolution and map a larger area. The final result, however, is a bit disappointing in the way it has been analyzed so far. On the field scale the mean flux rates match, but there is hardly any correlation between the measurements at the local points using the new system and using chamber systems. A reasonable fit of mean values could also be expected for the same chamber measurements and a an traditional EC system mounted at a 10 m high tower , so that the foot print more or less matches ( although the footprint depends on wind direction etc). I miss details on how the measurements and calculations are really done. They look a bit different from tradional EC calculation – what does it mean? Is the vertical wind speed used in both directions in the calculation, i. e. plus and minus? How was the background concentration determined? The authors mention that vertical wind flow was measured . . ...mounted either with the gas analysers (as I would expect in a modified EC approach) or at a fixed point in the field – how do the authors then combine the latter vertical wind speed at the fixed point to the changes in concentration somewhere else? Did the authors test if the vertical wind speeds or exchange rates were the same in the same moment all over the field?.

But I generally like the idea to develop and test this new idea, so I recommend to revise the manuscript, and add more details about the ideas and routines used for this new approach.

Specific comments:

Maybe the authors could explicitly define in the abstract and intro a name for their methods, like they implicitly did.

L33. The description about how chamber measurements work is correct, but I recommend to add a reference. Unfortunately there are different names for the systems, like open or closed chambers, steady state or non-steady state chamber etc. Open or closed loop seems clear as terminology , but I haven't read it yet, so it might add to this confusion.

L45 but a certain minimum turbulence is always need for the EC method.

L49, not 100% correct; a certain minimum turbulence is always need for the EC

method.

Eq.2 How do you know/ measure/ calculate the background

L114/15 please see my comments in the main comments

References: please check the formatting of $CO_2$ and $CH_4$ and other subscript letters

---

## Author Comment (AC1) · 27 Jul 2020

The authors would like to thanks the reviewers for taking the time to review this manuscript and for the provision of constructive comments that will no doubt improve the dissemination of this research. We have broken the reviewer comments down and provided a response to each.

Reviewer 1: "Nowadays, there are several research institutes/companies worldwide developing drone-based solutions, using commercial or homemade sensor technologies, including open-path lasers, directed lasers, IR sensors. . . Some developments about this approach are welcome in the introduction, because, depending on the topography/vegetation, drone flights can also be operated at short distances from the ground, offering a geographical coverage rate far higher than the one reached by systems operated at walking speed. The reference Feitz et al. (2018) is mentioned but not as a basis of comparison e.g. for evaluating positive aspects and drawbacks". We agree that the inclusion of previous research on UAV detection of soil gas flux would be of benefit to include and will modify the manuscript accordingly. Although UAV detection methods can provide geographical coverage rates far in excess of walking speed for soil gas flux measurements, there are downsides related to the density of measurements and the height at which measurements are collected. UAVs have weight and power limitations, which raises the question of whether smaller, low power sensors would have the sensitivity and low noise levels required to observe small anomalies at flight height. For example, Feitz et al. (2018) used a small NDIR CO2 sensor having a relatively high resolution of 10 ppm, with a maximum flight time of only 15 minutes. In this study the release rate was also relatively high, with the 50 g CO2 / minute released at a single point at 30cm height being equivalent to a 3000 g/m2 d soil gas leak over an area of 25 m2 or a 700 g/m2 d leak over 100m2; note that flux values above 500 g/m2 d tend to kill vegetation and thus leave an obvious impact. Copter style UAVs will also induce down-wash, which will cause air mixing and dilution requiring the sensor to be slung below. Fixed wing models will not have this problem, however their flight speed will be greater; faster motion will require extremely fast sensor response times (almost instantaneous) while maintaining a stable, sensitive measurement. Another important issue is what is considered a safe (or legal) flight height, as flying and measuring above the 10cm height used here will result in more dilution and greater horizontal wind speeds. Considering these issues it seems that drones may have potential for locating large leaks, but may not be adaptable to quantification of lower levels as discussed here.

Reviewer 1: "One of these assumptions is the absence of horizontal wind at land surface or at list little horizontal wind flow. More developments on this point are required to assess the potential use of the proposed method, and also on the related parameters such as soil roughness (e.g. Giannico et al., 2018. Contributions of landscape

heterogeneity within the footprint of eddy-covariance towers to flux measurements. https://doi.org/10.1016/j.agrformet.2018.06.004)". We agree with the reviewer, more research is required to assess the potential use of the field-scale method under different atmospheric and field conditions. The method presented within the manuscript is limited to our initial findings and we encourage others to expand on these findings and assess the technique for robustness under different settings.

Reviewer 1: "First, literature data are welcomed to evaluate the soundness of this major hypothesis". We will expand the literature review and further compare and contrast the method to similar methods such as the use of Eddy Covariance. This should help readers to evaluate the soundness of the hypothesis presented in the manuscript.

Reviewer 1: "Second, data related to the acquisitions performed in Italy and UK have to be presented: there is no Figure showing the wind conditions during acquisitions and this information is lacking. This may partly explain some of the differences reported in Figure 1 (Italian site) and probably some of the differences observed at the UK site (data not shown but this can be deduced from Figure 3)." We will add a figure showing the wind conditions during acquisitions and discuss impact on results. Horizontal wind flow was minimal during both data collection campaigns, but there were differences in the vertical wind component.

Reviewer 1: "On the contrary, are there some information on the diurnal variability of $CO_2$ emissions at the UK site that can explain, at least partly, the poor agreement chamber measurements and measurements using the authors' approach (chamber data were probably acquired over a longer period than the other data)?" The UK soil-gas data collected via the chamber method was collected over several hours and may show some diurnal variability. We will use the timestamps from both the field-scale and chamber methods to analyse the diurnal variability of $CO_2$ emissions at the UK site. We will add the findings to a revised manuscript.

Reviewer 1: "Do the authors think the "open-field scale" approach can be used with a

sufficient degree of confidence at the UK site or would they prefer using the chamber measurements"? The amount of confidence in the field-scale technique would depend on the requirements of any particular study and in the atmospheric conditions. If detailed fluxes were required over a small area, then the chamber technique would be preferable as there is no atmospheric dilution/mixing. However, if average flux was required over a larger area the field-scale method would be preferable. This method is not designed to replace chamber methods, but to compliment them whilst removing some of the limitations on measuring soil-gas fluxes over larger areas(100m2+).

Reviewer 1: "Back on the "absence of horizontal wind" assumption: the authors mention potential applications of their method, including leakage detection. What about leakage detection when there is little to no vegetation on the ground (the CO2FieldLab experiment is mentioned in the references)? Does the assumption seem realistic in that case? What about using such a method in desert environments? The authors also mention the Weyburn case: what could be the influence of frozen conditions on wind conditions close to ground surface"? There is no reason why landuse should impact in the ability to derive fluxes using this method, as changes to vertical wind flow generated by landuse are taken into account through the anemometer observations. There may be some limitations on use in certain conditions that generate no vertical upwards wind flow. However, these areas should also facilitate mounting the sensors closer to the ground, thus reducing horizontal flow. Regarding the "absence of horizontal wind" assumption, low horizontal windspeeds should not impact the average derived flux if the area of measurement is sufficiently large. Comparatively, horizontal wind flow should be greater in areas with no or little vegetation due to reductions in drag, however, they would also allow the sensors to be mounted lower to the ground (which would reduce horizontal windspeed). Further research would be required to assess the field area to windspeed relationship in deriving an average flux.

Reviewer 1: "The acquisitions were performed with a sensor mounted at 10 cm from ground surface: if the vegetation is higher than 10 cm and not grazed or mown

(spring/summer conditions), how can be the method adapted?" The method could be adapted in several ways, but more research will be needed to quantify the impacts. For example, the measurement height could be raised above the vegetation.

Reviewer 1: "A picture showing/describing the "open-field scale" system in field-use conditions is lacking. It is always informative to have such Figure." We agree with the reviewer and will add such an image to the revised manuscript.

Reviewer 1: "It would have been interesting to compare with a third approach, intrinsically related to the "open-field scale" approach: the "traditional" EC monitoring. Why has this not been performed? It would have given interesting information on the benefits of the "open-field scale" approach, e.g.: is the "open-field scale" approach offering the same smoothing of anomalies than the EC approach, or does it give a better rendering"? The method could be compared to multiple approaches (UAV, EC, EO, etc), however fieldwork had to be focused on specific goals within the project's budget, time and logistics. The comparison to chamber methods was made because this technique is currently used in UK for the assessment of soil gas flux at field-scales. The EC footprint method may be limited in this application by wind speed/direction and would be impractical to move to multiple locations in order to cover several fields. We agree that a comparison between multiple soilgas flux methods would be useful and should form the basis of further research.

Reviewer 1: "On the discrepancies between chamber data and "open-field scale" approach (especially for the UK site): a Figure comparing the results of the two methods is missing and this is needed; Figure 3 is not sufficient because we only see a point cloud". To allow improved assessment of the technique we will add a figure with individual points plotted against each other. Due to atmospheric dilution, we would not expect individual points to be consistent. Indeed, the field-scale method was not designed to be consistent with chamber measurements at individual points, but to derive comparable average fluxes.

Reviewer 1: "There is nearly no discussion on the differences between the two approaches for the UK site (only line 159). There is a lack of comparison with literature data, even if the "open-field scale" approach is not described in there because of its novelty. For example, what can be the differences with the use of vehicle-mounted (or walking use) of open-path lasers without quantifying the vertical wind flows"? We will expand on the discussion of differences between approaches for the UK site and for the comparison through literature as suggested. Without determining the vertical wind flows, a standard survey with a vehicle or walked open path laser can only address the mapping of measured values and potential location of flux anomalies, however it cannot quantify that flux.

Reviewer 1: "Because the "open-field scale" approach, like the EC approach, is supposed to give smoothed information on extreme values, what is the monitoring approach suggested by the authors in case there is a need to quantify these very high values. Use of "open-field scale" approach first and then perform chamber measurements, e.g. as suggested by Eugster and Merbold, 2015 (Eddy covariance for quantifying trace gas fluxes from soils. SOIL 1, 187–205. https://doi.org/10.5194/soil-1-187-2015)"? As suggested by the reviewer, where elevated values are detected using the field-scale approach, follow-up measurements would be undertaken using chamber methods to better quantify the source and scale of extreme values. Using this combined approach should be much quicker and less labour intensive than relying on chamber techniques alone.

Reviewer 2: "On the field scale the mean flux rates match, but there is hardly any correlation between the measurements at the local points using the new system and using chamber systems". This result was expected as even a little horizontal wind flow makes the measurement spatially non-coherent with the surface below the measurement point. The point of the field-scale approach is to identify elevated soil-gas fluxes over large areas. Chamber methods could then be used to pin-point and quantify extreme values where necessary.

Reviewer 2: "A reasonable fit of mean values could also be expected for the same chamber measurements and a an traditional EC system mounted at a 10 m high tower, so that the foot print more or less matches ( although the footprint depends on wind direction etc)". As noted in the response to Reviewer 1, the EC footprint method may be limited in this application by wind speed/direction and would be impractical to move to multiple locations in order to cover several fields. In addition, EC measurements are impacted by barriers, like the forests that border some of the study fields, and do not give a spatial distribution of anomalies like the present technique.

Reviewer 2: "I miss details on how the measurements and calculations are really done. They look a bit different from traditional EC calculation – what does it mean? Is the vertical wind speed used in both directions in the calculation, i. e. plus and minus? How was the background concentration determined"? We will expand on the details of how measurements and calculations were undertaken including differences to traditional EC calculation and what this means from a theoretical standpoint. It would not be possible to use this technique under a negative vertical windspeed (i.e., towards the ground), as the sensor would be measuring atmospheric $CO_2$ instead of that from the soil. Background concentrations were derived by assessing $CO_2$ concentrations from across the site against and calculating concentration minima with outliers removed.

Reviewer 2: "The authors mention that vertical wind flow was measured . . ...mounted either with the gas analysers (as I would expect in a modified EC approach) or at a fixed point in the field – how do the authors then combine the latter vertical wind speed at the fixed point to the changes in concentration somewhere else"? Where a fixed central location was used to measure the vertical windspeed, we assumed this to be spatially uniform across the field site. A better approach is to mount the sensor to the gas analyser, but this is not always possible given the power requirements of the instruments.

Reviewer 2: "Did the authors test if the vertical wind speeds or exchange rates were the same in the same moment all over the field"? We did not, however the surface

characteristics (slope, vegetation, soil type, etc) of the fields were reasonably uniform. Mounting the sensor near to the gas analyser on the cart would be the preferred approach as spatial variation in vertical windflow would be measured.

Reviewer 2: "Maybe the authors could explicitly define in the abstract and intro a name for their methods, like they implicitly did". We will change the manuscript and explicitly call this approach the "field-scale soil gas flux" technique.

Reviewer 2: "The description about how chamber measurements work is correct, but I recommend to add a reference". We agree and will provide further references on chamber techniques.

---

## Author Response (AR2)

The authors would like to thanks the reviewers for taking the time to review this manuscript and provide constructive comments that will no doubt improve the presentation of this research. We have broken the reviewer comments down and provided a response to each.

**Reviewer 1 comment:** *"Nowadays, there are several research institutes/companies worldwide developing drone-based solutions, using commercial or homemade sensor technologies, including open-path lasers, directed lasers, IR sensors. . . Some developments about this approach are welcome in the introduction, because, depending on the topography/vegetation, drone flights can also be operated at short distances from the ground, offering a geographical coverage rate far higher than the one reached by systems operated at walking speed. The reference Feitz et al. (2018) is mentioned but not as a basis of comparison e.g. for evaluating positive aspects and drawbacks."*

Author response: We agree that the inclusion of previous research on UAV detection of soil gas flux would benefit the article's overview of state-of-the-art and thus we have modified the manuscript accordingly. We believe, however, that this approach presently has limitations; in the manuscript we have alluded to this briefly but expand on our thoughts here. Although UAV detection methods can provide geographical coverage rates far in excess of walking speed for soil gas flux measurements, there are downsides related to the density of measurements and the height at which measurements are collected. UAVs have weight and power limitations, which raises the question of whether smaller, low power sensors would have the sensitivity and low noise levels required to observe small and/or weak anomalies at flight height. For example, Feitz et al. (2018) used a small NDIR $CO_2$ sensor, having a relatively high resolution of 10 ppm, mounted on a UAV with a maximum flight time of only 15 minutes. In this study the release rate was also relatively high, with the 50 g $CO_2$ / minute released at a single point at 30cm height, which is equivalent to a 3000 g/m2 d soil gas leak over an area of 25 m2 or a 700 g/m2 d leak over 100m2. Note that flux values above 500 g/m2 d tend to kill vegetation and thus leave an obvious impact, meaning that monitoring techniques must be able to detect lower levels to be useful. Copter style UAVs will also induce down-wash, which will cause air mixing and dilution requiring the sensor to be slung below. Fixed wing models will not have this problem, however their flight speed will be greater; faster motion will require extremely fast sensor response times (almost instantaneous) while maintaining a stable, sensitive measurement. Another important issue is what is considered a safe (or legal) flight height, as flying and measuring above the 10cm height used here will result in more dilution and greater horizontal wind speeds. Considering these issues it seems that drones may have potential for locating large leaks, but may not be adaptable to quantification of lower levels as discussed here.

Manuscript changes: (Ln60-65) Added a section on UAV use and drawbacks for measuring soil-gas flux at the field-scale.

**Reviewer 1:** *"One of these assumptions is the absence of horizontal wind at land surface or at list little horizontal wind flow. More **developments on this point are required** to assess the potential use of the proposed method, and also on the related parameters such as soil roughness (e.g. Giannico et al., 2018. Contributions of landscape heterogeneity within the footprint of eddy-covariance towers to flux measurements. https://doi.org/10.1016/j.agrformet.2018.06.004)."*

Author response: We agree with the reviewer, more research is required to assess the potential use of the field-scale method under different atmospheric and field conditions. The method presented within the manuscript is limited to our initial findings and we encourage others to expand on these findings and assess the technique for robustness under different settings.

Manuscript changes: Acknowledged further research is required in Abstract and summary (Ln 210).

**Reviewer 1:** *"First, literature data are welcomed to evaluate the soundness of this major hypothesis."*

Author response: We have expanded the literature review and further compare and contrast the method to similar techniques such as Eddy Covariance. This should help readers to evaluate the soundness of the hypothesis presented in the manuscript.

Manuscript changes: Expanded the literature review (Section 1) and added a table (Table 1) to compare and contrast the open-field method to EC and chambers.

**Reviewer 1:** *"Second, data related to the acquisitions performed in Italy and UK have to be presented: there is no Figure showing the wind conditions during acquisitions and this information is lacking. This may partly explain some of the differences reported in Figure 1 (Italian site) and probably some of the differences observed at the UK site (data not shown but this can be deduced from Figure 3)."*

Author response: We have assessed wind speed and direction against difference in flux between chamber and the open-field technique. Analysis of the results found no discernible link between wind conditions and differences in flux. However, it may be that other site-specific or external influences are masking this relationship if it exists. We have not included this analysis in the revised manuscript as it was felt that it did not add to the presented method and may dissuade other researchers from undertaking similar analysis under more suitable conditions (ie., designing specific lab/field experiments to quantitatively assess this relationship). Instead it has been included in the supplementary material.

Manuscript changes: Assessment added to supplementary material

**Reviewer 1:** *"On the contrary, are there some information on the diurnal variability of CO2 emissions at the UK site that can explain, at least partly, the poor agreement chamber measurements and measurements using the authors' approach (chamber data were probably acquired over a longer period than the other data)?"*

Author response: The UK soil-gas data collected via the chamber method was collected over several hours and may show some diurnal variability. As with the relationship between wind speed and flux, no relationship was found between diurnal variability and difference in flux. For the same reason we have decided not to include this is our revised manuscript.

Manuscript changes: None

**Reviewer 1:** *"Do the authors think the "open-field scale" approach can be used with a sufficient degree of confidence at the UK site or would they prefer using the chamber measurements?"*

Author response: The amount of confidence in the field-scale technique would depend on the requirements of any particular study and the atmospheric conditions. If detailed fluxes were required over a small area, then the chamber technique would be preferable as there is no atmospheric dilution/mixing. However, if average flux was required over a larger area the field-scale method would be preferable. This method is not designed to replace chamber methods, but to compliment them whilst removing some of the limitations on measuring soil-gas fluxes over larger areas(>100m$^2$).

Manuscript changes: None

**Reviewer 1:** *"Back on the "absence of horizontal wind" assumption: the authors mention potential applications of their method, including leakage detection. What about leakage detection when there is little to no vegetation on the ground (the CO2FieldLab experiment is mentioned in the references)? Does the assumption seem realistic in that case? What about using such a method in desert environments? The authors also mention the Weyburn case: what could be the influence of frozen conditions on wind conditions close to ground surface?"*

Author response: There is no reason why land use should impact in the ability to derive fluxes using this method, as changes to vertical wind flow generated by land use are taken into account through the anemometer observations. Regarding the "absence of horizontal wind" assumption, low horizontal wind speeds should not impact the average derived flux if the area of measurement is sufficiently large. Comparatively, horizontal wind flow should be greater in areas with no or little vegetation due to reductions in drag, however, they would also allow the sensors to be mounted lower to the ground (where horizontal wind speed is reduced). Further research would be required to assess the field area to wind speed relationship in deriving an average flux.

Manuscript changes: None

**Reviewer 1:** *"The acquisitions were performed with a sensor mounted at 10 cm from ground surface: if the vegetation is higher than 10 cm and not grazed or mown (spring/summer conditions), how can be the method adapted?"*

Author response: The method could be adapted in several ways, but more research will be needed to quantify the impacts. For example, the measurement height could be raised above the vegetation.

Manuscript changes: None

**Reviewer 1:** *"A picture showing/describing the "open-field scale" system in field-use conditions is lacking. It is always informative to have such Figure."*

Author response: Due to COVID restrictions we are unable to take new photographs of the open-field system in use before the revision deadline. We used a photograph of the prototype system from a past campaign.

Manuscript changes: Added Figure 1 to manuscript.

**Reviewer 1:** *"It would have been interesting to compare with a third approach, intrinsically related to the "open-field scale" approach: the "traditional" EC monitoring. Why has this not been performed? It would have given interesting information on the benefits of the "open-field scale" approach, e.g.: is the "open-field scale" approach offering the same smoothing of anomalies than the EC approach, or does it give a better rendering?"*

Author response: The method could be compared to multiple approaches (UAV, EC, EO, etc), however fieldwork had to be focused on specific goals within the project's budget, time and logistics. The comparison to chamber methods was made because this technique is currently used in the UK for the field-scale assessment of soil gas flux . The EC footprint method may be limited in this application by wind speed/direction and would be impractical to move to multiple locations in order to cover several fields. We agree that a comparison between multiple soilgas flux methods would be useful and should form the basis of further research.

Manuscript changes: Expanded the literature review (Section 1) and added a table (Table 1) to compare and contrast the open-field method to EC and chambers.

**Reviewer 1:** *"On the discrepancies between chamber data and "open-field scale" approach (especially for the UK site): a Figure comparing the results of the two methods is missing and this is needed; Figure 3 is not sufficient because we only see a point cloud."*

Author response: To allow improved assessment of the technique we have added a figure with individual points plotted against each other. Due to atmospheric dilution, we would not expect individual points to be consistent. Indeed, the field-scale method was not designed to be consistent with chamber measurements at individual points, but to derive comparable average fluxes.

Manuscript changes: Added figure 3 to show discrepancies between chamber data and open-field scale approach at each chamber measurement point.

**Reviewer 1:** *"There is nearly no discussion on the differences between the two approaches for the UK site (only line 159). There is a lack of comparison with literature data, even if the "open-field scale" approach is not described in there because of its novelty. For example, what can be the differences with the use of vehicle-mounted (or walking use) of open-path lasers without quantifying the vertical wind flows?"*

Author response: The approach was the same for the both sites and the manuscript has been modified to include a comparison through literature as suggested. Without determining the vertical wind flows, a standard survey with a vehicle or walked open path laser can only address the mapping of measured values and potential location of flux anomalies, however it cannot quantify that flux.

Manuscript changes: Added figure 3 for better comparison of the data at the sites. Expanded the literature review (Section 1) and added a table (Table 1) to compare and contrast the open-field method to EC and chambers.

**Reviewer 1:** *"Because the "open-field scale" approach, like the EC approach, is supposed to give smoothed information on extreme values, what is the monitoring approach suggested by the authors in case there is a need to quantify these very high values. Use of "open-field scale" approach first and then perform chamber measurements, e.g. as suggested by Eugster and Merbold, 2015 (Eddy covariance for quantifying trace gas fluxes from soils. SOIL 1, 187–205. https://doi.org/10.5194/soil-1-187-2015)?"*

Author response: As suggested by the reviewer, where elevated values are detected using the field-scale approach, follow-up measurements would be undertaken using chamber methods to better quantify the source and scale of extreme values. Using this combined approach should be much quicker and less labour intensive than relying on chamber techniques alone.

Manuscript changes: Highlighted this combined approach in the abstract and in the summary.

**Reviewer 2:** *"On the field scale the mean flux rates match, but there is hardly any correlation between the measurements at the local points using the new system and using chamber systems."*

Author response: This result was expected as even a little horizontal wind flow makes the measurement spatially non-coherent with the surface below the measurement point. The point of

the field-scale approach is to identify elevated soil-gas fluxes over large areas. Chamber methods could then be used to pin-point and quantify extreme values where necessary.

Manuscript changes: None

**Reviewer 2:** *"A reasonable fit of mean values could also be expected for the same chamber measurements and a an traditional EC system mounted at a 10 m high tower, so that the foot print more or less matches ( although the footprint depends on wind direction etc)."*

Author response: As noted in the response to Reviewer 1, the EC footprint method may be limited in this application by wind speed/direction and would be impractical to move to multiple locations in order to cover several fields. In addition, EC measurements are impacted by barriers, like the forests that border some of the study fields, and do not give a spatial distribution of anomalies like the present technique.

Manuscript changes: Modified the literature review in the introduction to highlight this point.

**Reviewer 2:** *"I miss details on how the measurements and calculations are really done. They look a bit different from traditional EC calculation – what does it mean? Is the vertical wind speed used in both directions in the calculation, i. e. plus and minus? How was the background concentration determined?"*

Author response: We address details of how measurements and calculations were undertaken, including differences to traditional EC calculation and what this means from a theoretical standpoint. It would not be possible to use this technique under a negative vertical windspeed (i.e., towards the ground), as the sensor would be measuring atmospheric $CO_2$ instead of that from the soil. Background concentrations were derived by assessing $CO_2$ concentrations from across the site and calculating concentration minima with outliers removed.

Manuscript changes: Modified the literature review, experimental theory (Ln 95) and changed measurement and post-processing sections to expand on open-field approach measurements, processing and theory.

**Reviewer 2:** *"The authors mention that vertical wind flow was measured . . ...mounted either with the gas analysers (as I would expect in a modified EC approach) or at a fixed point in the field – how do the authors then combine the latter vertical wind speed at the fixed point to the changes in concentration somewhere else?"*

Author response: Where a fixed central location was used to measure the vertical windspeed, we assumed this to be spatially uniform across the field site. A better approach is to mount the sensor to the gas analyser, but this is not always possible given the power requirements of the instruments.

Manuscript changes: None

**Reviewer 2:** *"Did the authors test if the vertical wind speeds or exchange rates were the same in the same moment all over the field?*

Author response: We did not, however the surface characteristics (slope, vegetation, soil type, etc) of the fields were reasonably uniform. Mounting the sensor near to the gas analyser on the cart would be the preferred approach as spatial variation in vertical wind flow would be measured.

Manuscript changes: None

**Reviewer 2:** *"Maybe the authors could explicitly define in the abstract and intro a name for their methods, like they implicitly did."*

Author response: We agree with the suggestion.

Manuscript changes: Changed the name of the technique and explicitly named it in the abstract and introduction.

**Reviewer 2:** *"The description about how chamber measurements work is correct, but I recommend to add a reference."*

Author response: We agree.

Manuscript changes: Added references to chamber literature into Section 1.

[revised manuscript text omitted]

---

## Author Response (AR3)

Dear GI,

We have modified the data availability statement in line with the Editor recommendations and GI guidelines.

Kind Regards,

Andy